# LLS: Regulating Neural Network Training via Learnable Label Smoothing

## Abstract

Training a neural network using one-hot targets often leads to the issue of overconfidence. To address this, Label Smoothing has been introduced, modifying the targets to a mix of one-hot encoding and a uniform probability vector. However, the uniform probability vector indiscriminately assigns equal weights to all categories, thereby undermining inter-category relationships. To overcome these challenges, we propose a novel solution, Learnable Label Smoothing (LLS), that aims to regulate training by granting networks the ability to assign optimal targets. Unlike conventional methods, Learnable Label Smoothing utilizes probability vectors unique to each category, resulting in diverse targets. The acquired relationships are beneficial for regularization and also prove to be transferable, facilitating knowledge distillation even in the absence of a Teacher model. Our extensive experiments across multiple datasets highlight the advantages of our method in addressing both overfitting and the preservation of inter-category relationships in neural network training.

## 1 Introduction

The traditional method of training neural networks involves the utilization of one-hot targets and cross-entropy loss, a long-standing practice in the field. However, the use of one-hot targets has been recognized for its tendency to instigate overconfidence within the network, potentially hampering its generalization capabilities Szegedy et al. (2016). Over the years, various regularization techniques, such as Cutout (Devries & Taylor (2017)), Mixup (Zhang et al. (2018)), CutMix (Yun et al. (2019)), and others (Hendrycks et al. (2020); Gong et al. (2021)), have been introduced to address this issue, often involving modifications to the input data. An alternative strategy is Label Smoothing, which adjusts target labels during training by adding a uniform label distribution over the categories to the one-hot target (Szegedy et al. (2016)). Training with Label Smoothing has proven effective in enhancing generalization and has been widely adopted.

Despite the advantages of Label Smoothing, it is known to disrupt the relationships between categories (Müller et al. (2019)). This problem arises from the use of a uniform probability vector in generating smoothed targets, assigning equal importance to all negative categories. Consequently, the network is instructed to treat all categories as equally distinct from each other, leading to compact and equidistant category clusters in the feature space (Müller et al. (2019)). This outcome is undesirable; for e.g., targets for the *Dog* class should have a relatively higher similarity with the *Cat* class, as compared to the *Truck* class. Enforcing uniform inter-category relationships limits the model's performance Zhang et al. (2021). Inter-category relationship is crucial for applications such as Knowledge Distillation, dealing with missing data, and learning from noisy labels (Hinton et al. (2015); Müller et al. (2019); Zhang et al. (2021)). This prompts two fundamental questions: (1) Is it possible to regulate confidence while preserving the inter-category relationship? and, (2) What alternative should be employed in place of the uniform probability vector?

This paper introduces a novel solution, termed *Learnable Label Smoothing* (LLS), to address these questions. Our approach aims to train the network to learn the optimal target vector, as illustrated in Figure 1. We propose a category-wise learnable probability vector. By combining these probability vectors with the one-hot labels, similar to Label Smoothing, we create targets unique to each category. For a dataset with $K$ categories, these category-wise learnable probability vectors together form the $K \times K$ $Q$-matrix, whose rows $Q_k$ encode the inter-category similarities.

What should be the target label
for the below image?

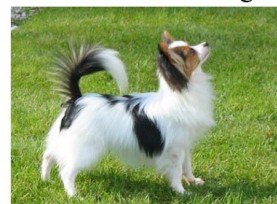 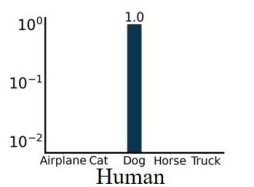 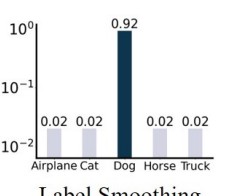 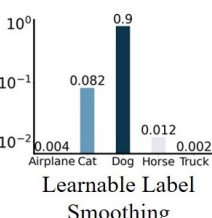

Figure 1: Toy Diagram. Our method seeks to regulate training by empowering the network to determine its optimal targets.

We demonstrate empirically that Learnable Label Smoothing outperforms Label Smoothing and its other variations. Furthermore, networks trained with Learnable Label Smoothing prove to be more effective Teacher models for Knowledge Distillation. The learned $Q$-Matrices enable seamless knowledge transfer and distillation even in the absence of the Teacher network. A $Q$-Matrix learned from a large dataset can be used to regularize its subsets (category-wise and sample-wise) of the data and reduces the necessity for frequent relearning of the $Q$-Matrix. These characteristics enhance the $Q$-Matrix's versatility and widen the scope of Learnable Label Smoothing's potential applications.

## 2 RELATED WORK

Training neural networks with 1-hot targets are well-known for inducing overconfidence and adversely affecting generalization (Szegedy et al. (2016)). Numerous regularization techniques have been proposed to mitigate this issue, with a predominant focus on enhancing input data (Zhang et al. (2018); Yun et al. (2019); Devries & Taylor (2017)). Label regularization techniques seek to modify targets to alleviate overconfidence. Label smoothing is one of the straightforward solutions that mix the 1-hot vector with a uniform vector, weighted by a hyper-parameter $\alpha$ (Szegedy et al. (2016)). Despite its merits, Label Smoothing has the drawback of disrupting inter-category relationships by assigning equal weights to all negative categories (Müller et al. (2019)). Our novel approach diverges from a uniform vector, opting instead to learn the probability vector for mixing to prevent disrupting inter-category relationships.

Entropy maximization on network predictions emerges as an alternative to Label Smoothing (Pereyra et al. (2017)). This technique provides greater flexibility to samples, allowing them to determine the weight of negative categories instead of adhering to uniform weights. Our approach leverages entropy maximization loss on network predictions and trains the network to learn the targets. Focal loss was proposed as a modification of the cross-entropy loss function (Mukhoti et al. (2020); Lin et al. (2017)). It allocates higher weights to samples with low confidence and lower weights to those with high confidence. This loss works by minimizing a regularized KL divergence and preventing the model from becoming excessively overconfident. This further underscores our selection of entropy maximization in regulating targets.

Knowledge Distillation is recognized as a form of label regularization (Hinton et al. (2015); Yuan et al. (2020)). It involves producing targets from a larger network (the Teacher) and passing this knowledge to a smaller network (the Student) on a per-sample basis. The relationship of each sample to negative categories, as learned by the teacher, aids in regulating the Student networks (Hinton et al. (2015)). In line with this concept, a trained network was employed to train another (same architecture) network in Teacher-Free Knowledge Distillation (Yuan et al. (2020)). However, this approach incurs significant computational expenses as it necessitates training a network twice and generating outputs using online training. An alternative, Teacher-Free regularization, behaves similarly to Label Smoothing but utilizes a high mixing coefficient of 0.9 to generate a smoothened probability vector (Yuan et al. (2020)). The network is trained to align predicted probabilities with this vector at a high temperature, reducing computational costs but still relying on a uniform vector. Our method departs from a uniform probability vector when generating a regularized target. Online Label Smoothing is another approach based on network predictions (Zhang et al. (2021)). It com-

putes average network predictions for each category and mixes them with a 1-hot probability vector. While it diminishes the need to train the network twice, it still carries a substantial computational overhead as average network predictions must be computed every epoch on the training set. Also, if the predictions become close to 1-hot, it results in training vectors to become 1-hot.

## 3 METHOD

### 3.1 PRELIMINARIES

Let $D$ be a dataset with image label pairs $\{x, y\}$ where $x$ represents an image, and $y \in \{1, \ldots, K\}$ is the ground truth label. The ground truth labels are also represented as 1-hot vectors $p = [p_1, \ldots, p_K]^\top$, where $p_i \in \{0, 1\}$. Correspondingly, $p_i = 1$ when index $i = y$, else it is 0. The neural network with parameters $\theta$ is represented as $f_\theta(.)$. For a sample $x$, the output probability vector is denoted by $\hat{p} = f_\theta(x)$. The standard cross-entropy objective $H(p, \hat{p})$ is minimized for network training, and is computed as,

$$H(p, \hat{p}) = -p \log \hat{p} = -\sum_{i=1}^{K} p_i \log \hat{p}_i = -\log \hat{p}_y. \tag{1}$$

However, the conventional training approach utilizing a 1-hot vector is known to induce overconfidence (Szegedy et al. (2016)) and lead to poor calibration and over-fitting (Mukhoti et al. (2020); Lin et al. (2017)). To address this issue, Label Smoothing introduces a regularization technique by creating a modified target $p^{ls}$ (Szegedy et al. (2016)). This is achieved by combing the 1-hot probability $p$ with a uniform probability vector $u = [\frac{1}{K}, \ldots, \frac{1}{K}]^\top$, resulting in,

$$p^{ls} = (1 - \alpha)p + \alpha u. \tag{2}$$

Here, $\alpha$ is the smoothing hyper-parameter, typically set to 0.1. The network trained using the cross-entropy with the modified targets ($p^{ls}$), mitigates the problem as,

$$\begin{aligned} H(p^{ls}, \hat{p}) &= -p^{ls} \log \hat{p} \\ &= (1 - \alpha)H(p, \hat{p}) + \alpha H(u, \hat{p}) \\ &= (1 - \alpha)H(p, \hat{p}) + \alpha KL(u||\hat{p}) + \alpha H(u). \end{aligned} \tag{3}$$

Here, the first term is the cross-entropy between $H(p, \hat{p})$ scaled by $(1 - \alpha)$. The second term is the Kullback-Leibler Divergence between $u$ and $\hat{p}$ driving the predictions to become more uniform and reducing the confidence of predictions. The last term is the entropy over $u$, where $H(u) = -\sum_i u_i \log u_i$, which is a constant.

### 3.2 LEARNABLE LABEL SMOOTHING (LLS)

Our approach proposes to replace the uniform vector $u$ in Label Smoothing with a learnable probability vector, granting the network the ability to select optimal targets. Our learned target vector is of the form, $p^{lls} = (1 - \alpha) * p + \alpha * q$ where, $q$ is learned through network training. We argue that a 1-hot target vector is an overconfident and hard assignment of the image category. Label Smoothing ameliorates the effect of overconfidence by assuming a uniform prior label distribution. However, Label Smoothing could introduce unwanted biases when uniformly smoothing the probabilities (Lienen & Hüllermeier (2021)). We propose to learn the distribution $q$ and estimate the 'moving' target label $p^{lls}$ even as the network trains to align the prediction $\hat{p}$ with $p^{lls}$. We share a probability vector $q$ between all samples within a category, due to their shared relationships with other categories and employ distinct $q$ for each category. Hence, we learn a matrix $Q$ of dimensions $K \times K$, where row $i$ signifies a learnable probability vector $Q_i = [q_{i1}, q_{i2}, \ldots, q_{iK}]$ for category $i$. For a training sample $(x, y)$, the modified label is given as $p^{lls}$ where,

$$p^{lls} = (1 - \alpha) * p + \alpha * Q_y, \tag{4}$$

with $p$ as the 1-hot vector corresponding to ground truth label $y$, and $Q_y$ the $y$-th row of the learned $Q$ matrix. $\alpha$ is the hyper-parameter similar to Label Smoothing. The purpose of the $Q$-Matrix is to facilitate the acquisition of the optimal mixing probability vectors during Label Smoothing. We refer to our framework as *Learnable Label Smoothing* (LLS).

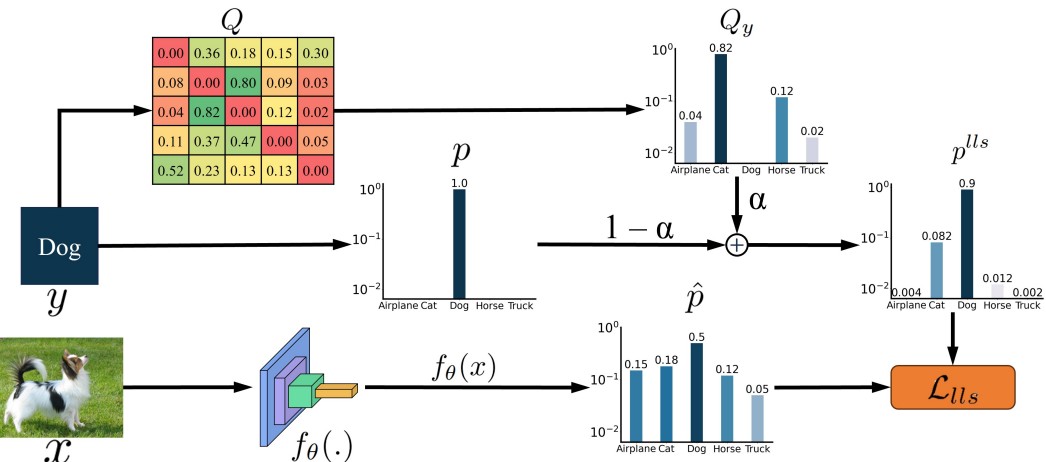

Figure 2: An overview of the proposed approach. Our approach utilizes a matrix $Q$ with dimensions $K \times K$ that serves as the repository for learnable probability vectors for each category. A given 1-hot vector $p$ of a category is mixed with its associated probability vector $Q_y$ from matrix $Q$, governed by the hyperparameter $\alpha$. This operation results in the target $p^{lls}$ which is used for training with $\mathcal{L}_{lls}$ loss.

### 3.3 TRAINING USING LLS

Given the Learnable Label Smoothing (LLS) target probability vector $p^{lls}$ and the network prediction $\hat{p}$, the standard training objective is the minimization of the cross-entropy loss $H(p^{lls}, \hat{p}) = -\sum_i p_i^{lls} \log \hat{p}_i$. The cross-entropy is an upper-bound on the KL-divergence between $p^{lls}$ and $\hat{p}$, where $H(p^{lls}, \hat{p}) = KL(p^{lls}||\hat{p}) + H(p^{lls})$. The second term $H(p^{lls})$ is the entropy of $p^{lls}$ which is 0 when $p^{lls}$ is 1-hot. When $p^{lls}$ is not 1-hot, minimizing cross-entropy $H(p^{lls}, \hat{p})$ also minimizes the entropy of $H(p^{lls})$, making $p^{lls}$ more 1-hot. This does not serve our purpose where we aim to retain the inter-category relationships in the target label. We propose to instead directly minimize the KL-divergence objective $KL(p^{lls}||\hat{p})$.

We term $KL(p^{lls}||\hat{p})$ as the Forward-KL. In standard Forward-KL divergence objectives, for e.g., $KL(r||s)$, the distribution $r$ is fixed, and $s$ is optimized to align with $r$. With $KL(p^{lls}||\hat{p})$, we have the challenge of a moving target where $p^{lls}$ is being learned as $\hat{p}$ aligns with it. That means we need both $\hat{p}$ and $p^{lls}$ need to be optimized to align with each other, respectively. However, Forward-KL produces disproportionate updates to the $\hat{p}$ and $p^{lls}$. This is mitigated when we also have a Reverse-KL term $KL(\hat{p}||p^{lls})$, which provides symmetry to the training loss function and ensures the target $p^{lls}$ and predictions $\hat{p}$ are updated with equal emphasis. We discuss this more using the derivatives of the Forward-KL and Reverse-KL in the Appendix (Section B). We also showcase the impact of not including Reverse-KL in the ablation study (Section 5.1 and Appendix Section H). The objective for training using the LLS is the sum of Forward-KL and Reverse-KL objectives.

$$\mathcal{L}_{lls} = KL(p^{lls}, \hat{p}) + KL(\hat{p}, p^{lls}) \tag{5}$$
$$= p^{lls} \log p^{lls} - p^{lls} \log \hat{p} + \hat{p} \log \hat{p} - \hat{p} \log p^{lls}$$
$$= -H(p^{lls}) + H(p^{lls}, \hat{p}) - H(\hat{p}) + H(\hat{p}, p^{lls}).$$

The first term $-H(p^{lls})$ is the negative entropy of the target, which, when minimized, drives the target $p^{lls}$ towards a uniform distribution. The target is $p^{lls} = (1 - \alpha)p + \alpha Q_y$, where only $Q_y$ varies. Minimizing $-H(p^{lls})$ effectively drives $Q_y$ to estimate inter-category relationships as $Q_y$ becomes more uniform. Similarly, the third term $-H(\hat{p})$ is the negative entropy of the predictions, which, when minimized, drives the predictions $\hat{p}$ towards a uniform distribution. This plays the role of Label Smoothing, which penalizes overconfidence in the predictions and alleviates overfitting. We name the second term $-H(p^{lls}, \hat{p})$, Forward Cross-Entropy, which aligns distributions $p^{lls}$ and $\hat{p}$. Similarly, we name the 4th term $-H(\hat{p}, p^{lls})$ Reverse Cross-Entropy. Minimizing these terms aligns the target $p^{lls}$ with the predictions $\hat{p}$.

Table 1: Results on CUB-200 and Flowers-102 for fine-grain classification. MV2: MobileNetV2 and RX denote the ResNet network with X number of layers.

| Dataset | CUB-200 | | | | Flowers-102 | | | |
|---|---|---|---|---|---|---|---|---|
| Network | MV2 | R18 | R50 | R101 | MV2 | R18 | R50 | R101 |
| 1-Hot | 77.76 | 78.08 | 80.81 | 81.71 | 91.03 | 90.37 | 90.69 | 91.74 |
| LS (Szegedy et al. (2016)) | 78.67 | 78.56 | 81.89 | 82.62 | 91.94 | 90.50 | 92.42 | 92.73 |
| TF-KD$_{reg}$($Yuan et\ al.$ (2020)) | 77.64 | - | 80.96 | - | 91.95 | - | 91.30 | - |
| OLS (Zhang et al. (2021)) | **79.95** | - | 82.47 | - | 92.73 | - | 92.86 | - |
| LLS (Ours) | 79.84 | **78.86** | **82.91** | **83.48** | **93.02** | **91.02** | **93.64** | **92.89** |

## 3.4 THE $Q$-MATRIX

Minimizing $-H(p^{lls})$ maximizes the entropy of the target $p^{lls} = (1 - \alpha)p + \alpha Q_y$, where only $Q_y$ varies. The entropy of $p^{lls}$ can be increased only by reducing $Q_{yy}$ and increasing the other components of $Q_y$ because the $y$-th component $p_y^{lls}$ is greater than the other components by a fixed constant term $(1 - \alpha)$. This propels the network to set $Q_{yy} \to 0$ and assign that probability to the other categories, thereby identifying inter-category relationships. Consequently, the $Q$ matrix exhibits the lowest values at the diagonals and higher values for semantically closer categories. The $Q$-Matrix is generally asymmetric, as we found the relation of a category to another does not reciprocate the same way. For e.g., using Figure 4b, the Pullover category has the highest similarity with the Shirt category, but the Shirt gets a higher similarity with the T-shirt category than the Pullover.

We learn a $Q_y$ vector for every category. This results in a $K \times K$ $Q$-matrix where every row $Q_y$ models the similarities of category $y$ with the other categories. The similarities learned by the $Q$-Matrix allow for knowledge transfer between different networks, especially when a teacher model can't be employed (More details in 5.3). Similarly, a $Q$-Matrix learned from a large dataset can be used to transfer its knowledge to its subsets (category-wise and sample-wise) of the data, which reduces the necessity for frequent relearning of the $Q$-Matrix (More details in Appendix Section D). The LLS method is depicted in the model diagram in Figure 2.

## 4 EXPERIMENTS

### 4.1 DATASETS AND SETUP

We evaluated our methodology across diverse settings, encompassing small-scale objects, large-scale objects, and scenarios demanding fine-grained classification. In the realm of small-scale classification, we used FashionMNIST (Xiao et al. (2017)), CIFAR10 (Krizhevsky et al. (2009)), and SVHN (Netzer et al. (2011)) datasets. These datasets, with images sized at $32 \times 32$, offer both diversity and challenges with 10-way classifications. SVHN presents an intriguing challenge as digits lack prominent inter-category relationships. For large-scale classification, our evaluation extended to CIFAR100 (Krizhevsky et al. (2009)), Tiny-ImageNet, and ImageNet-100. Due to hardware constraints, we leveraged Tiny-ImageNet and ImageNet-100 [URL], both subsets of the original ImageNet dataset (Deng et al. (2009)). Tiny-ImageNet possesses 200 categories with $64 \times 64$ images, while ImageNet-100, featuring the original $224 \times 224$ image size, encompasses 100 categories. In the fine-grained classification domain, our experiments focused on distinguishing between various bird species using the CUB-200 dataset (Wah et al. (2011)), different types of flowers using the Flowers-102 dataset (Nilsback & Zisserman (2008)), and different animals using the Animals-10N dataset (Song et al. (2019)).

We evaluated our approach on these datasets using different networks that are mentioned in their respective tables. We store $Q$-matrix as logits which are converted to probabilities using Softmax. The $Q$-matrix is initialized with zeros, leading to a uniform distribution as the starting point. The hyper-parameter $\alpha$ is set to $0.1$ for all experiments but optimizing $\alpha$ can provide additional gains (Explored in Appendix Section C). Detailed training procedure, the pseudo-code, and the code are provided in Appendix Section I, Section A, and the supplementary, respectively.

Table 2: Results on CIFAR100 and Tiny-ImageNet datasets. RX denotes the ResNet network with X number of layers.

| | CIFAR100 | | | | Tiny-ImageNet | | |
|---|---|---|---|---|---|---|---|
| Method | R18 | R34 | R50 | R101 | R18 | R50 | R101 |
| 1-Hot | 75.87 | 79.38 | 78.79 | 79.66 | 63.20 | 67.47 | 67.93 |
| LS Szegedy et al. (2016) | 77.26 | 79.06 | 78.80 | 79.88 | 63.13 | 67.63 | 68.31 |
| FL-3 Mukhoti et al. (2020) | - | - | 77.25 | - | - | 50.31 | 62.97 |
| FLSD-53 Mukhoti et al. (2020) | - | - | 76.78 | - | - | 50.94 | 62.96 |
| TF-KD$_{self}$ Yuan et al. (2020) | 77.10 | - | - | - | - | 68.18 | - |
| TF-KD$_{reg}$ Yuan et al. (2020) | 77.36 | - | - | - | - | 67.92 | - |
| Zipf Liang et al. (2022) | 77.38 | 77.38 | - | - | 59.25 | - | - |
| OLS Zhang et al. (2021) | - | 79.96 | 79.35 | 80.34 | - | - | - |
| LLS (Ours) | **79.69** | **80.71** | **81.04** | **81.21** | **64.58** | **68.28** | **69.42** |

Table 3: Results on SVHN, CIFAR10, FashionMNIST (FMNIST), Animals10N and ImageNet-100.

| Dataset | SVHN | CIFAR10 | FMNIST | Animals10N | ImageNet-100 | |
|---|---|---|---|---|---|---|
| Network | LeNet | AlexNet | AlexNet | ResNet18 | R18 | R50 |
| 1-Hot | 89.40±0.03 | 79.98±0.17 | 80.94±0.22 | 85.00±0.11 | 81.72 | 83.96 |
| LS | 89.35±0.09 | 80.66±0.20 | 81.15±0.24 | 86.13±0.19 | 82.22 | 84.58 |
| TFKD$_{reg}$ | 89.42±0.31 | 80.78±0.17 | 81.38±0.24 | 85.99±0.10 | 82.44 | 84.72 |
| OLS | 89.19±0.43 | 80.71±0.28 | 81.21±0.30 | 86.35±0.38 | 82.56 | 84.71 |
| LLS | **89.51±0.15** | **80.88±0.04** | **81.56±0.23** | **86.69±0.23** | **82.72** | **84.90** |

## 4.2 RESULTS

We conduct a comprehensive comparison of our approach against prominent label regularization techniques, including Label Smoothing (Szegedy et al. (2016)), Focal Loss (Mukhoti et al. (2020)), Teacher-Free Knowledge Distillation (Yuan et al. (2020)), and Online Label Smoothing (OLS) (Zhang et al. (2021)). The results are detailed in Table 1, 2, and 3. When results were not available in the original paper, we indicated them with '-'. Notably, for Tables 3, baseline experiments were conducted by us using the same setup as ours. Our approach consistently outperforms the alternatives across all the cases. Our approach imposes minimal overhead while achieving superior performance. We analyze the computation overhead of Learnable label smoothing in the Appendix Section G.

## 4.3 $Q$-MATRIX

We present $Q$-Matrices for CIFAR-10, Animals-10N, SVHN, and CIFAR100 in Figure 3, showcasing their learned relationships. The $Q$-Matrix notably reveals distinct connections among the categories. For CIFAR-10, we can observe that LLS assigns high values to similar categories and low values to dissimilar categories showcasing the learned inter-category relationship. Similarly, for Animals-10N, which is a fine-grain classification dataset and has 5 pairs of confusing pairs, high values are assigned to the other animals of the pair in $Q$-Matrix, showcasing their strong relationships. Furthermore, we depict the confusion matrix for the test set of the FashionMNIST in Figure 4a. It reveals a pattern consistent with the $Q$-Matrix showcased in Figure 3b. For instance, Shirts frequently get misclassified as T-shirts, followed by pullovers and coats, owing to their close semantic ties in that order. This correlation serves as a useful tool for estimating prediction uncertainty. For example, when an image is misclassified as a T-shirt, there is a higher likelihood of it being a Shirt and a significantly lower chance of it being a Bag. We also show $Q$-Matrix for a larger number of classes (CIFAR100) in Figure 3d. We can observe the same behavior here. For e.g., the Maple Tree has a similarity of 0.45 with the Oak Tree, 0.2 with the Pine Tree and Willow Tree, 0.02 with

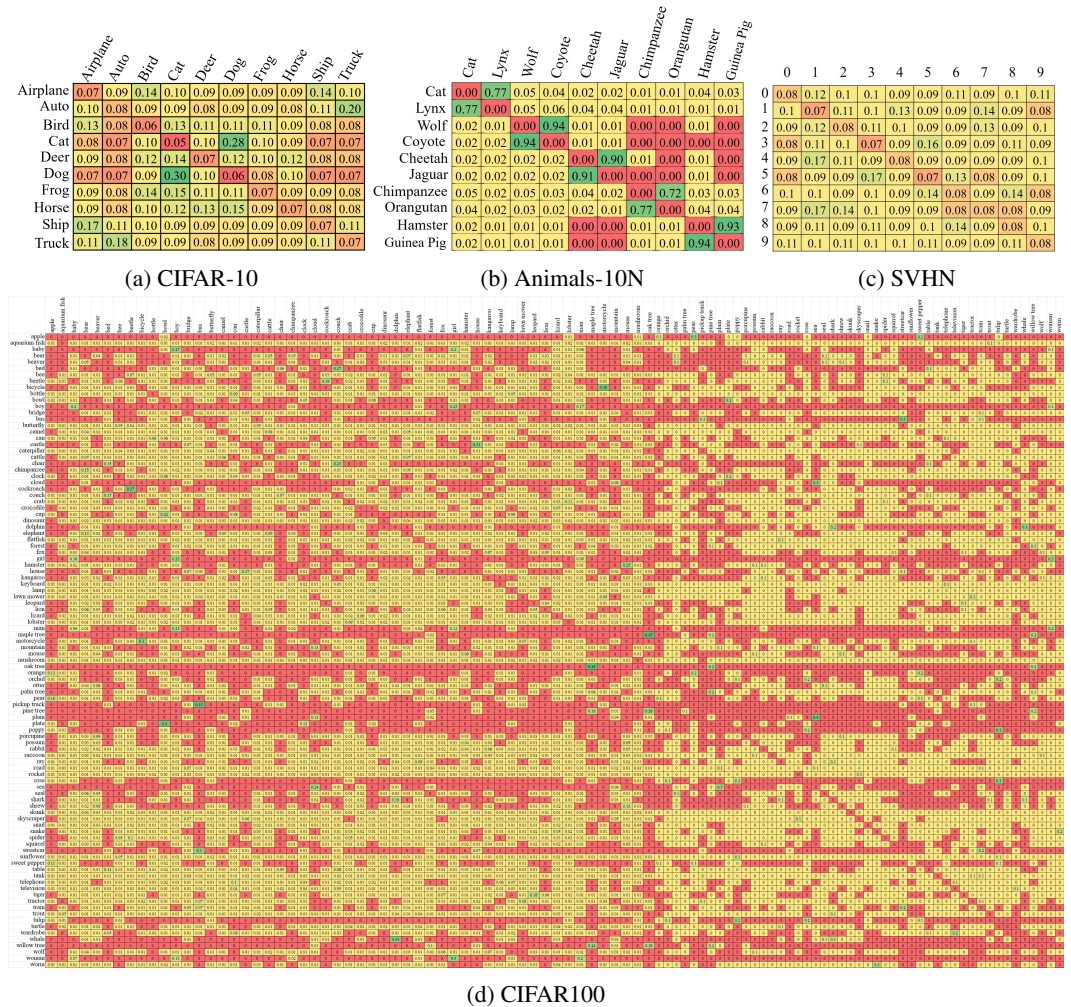

Figure 3: Learned $Q$-Matrices. We can observe $Q$-Matrix favors semantically closer categories. The final training label is obtained by mixing the $Q$-Matrix with the 1-hot vector of ground truth based on the $\alpha$ hyperparameter.

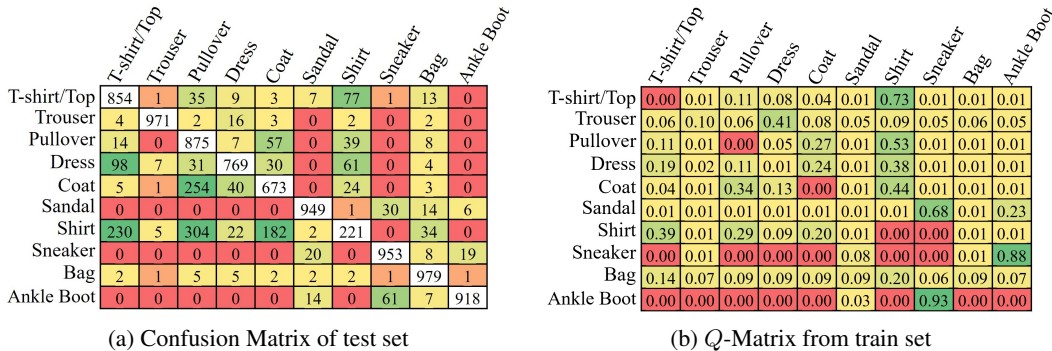

Figure 4: (a) Confusion Matrix on the validation set of FashionMNIST dataset and (b) Learned $Q$-Matrix from train set. We can observe misclassification in 4a follow the same trend as the relationship learned 4b.

the Forest, and very low with the rest. Another good example is the category Woman, which gets 0.3 similarity with Girl, 0.2 with Man, 0.13 with Boy, and 0.07 with Baby.

Table 4: Ablation Study experiments on Tiny-ImageNet and CUB-200 with ResNet-18 and ResNet-50. No FCE: No cross-entropy loss; No RCE: No reverse cross-entropy loss; No Pred EM: No entropy maximization loss on predictions; No Targets EM: No entropy maximization loss on targets; FCE only: Forward cross-entropy; Symmetric CE: Forward cross-entropy + Reverse cross-entropy.

| Description | Loss Terms | | | | Tiny-ImageNet | | CUB-200 | |
|---|---|---|---|---|---|---|---|---|
| | $H(p^{lls}, \hat{p})$ | $H(\hat{p}, p^{lls})$ | $-H(\hat{p})$ | $-H(p^{lls})$ | R18 | R50 | R18 | R50 |
| No FCE | ✗ | ✓ | ✓ | ✓ | 26.44 | 11.91 | 46.84 | 51.26 |
| No RCE | ✓ | ✗ | ✓ | ✓ | 64.14 | 68.04 | 78.84 | 82.88 |
| No Pred EM | ✓ | ✓ | ✗ | ✓ | 63.26 | 67.48 | 78.34 | 82.57 |
| No Targets EM | ✓ | ✓ | ✓ | ✗ | 63.80 | 67.51 | 78.10 | 82.78 |
| FCE only | ✓ | | | | 63.40 | 66.83 | 78.46 | 82.66 |
| Symmetric CE | ✓ | ✓ | ✗ | ✗ | 62.91 | 66.80 | 78.13 | 82.07 |
| Forward KL | ✓ | ✗ | ✗ | ✓ | 63.03 | 67.87 | 78.22 | 82.52 |
| Reverse KL | ✗ | ✓ | ✓ | ✗ | 26.58 | 14.40 | 46.62 | 53.56 |
| LLS | ✓ | ✓ | ✓ | ✓ | **64.58** | **68.28** | **78.86** | **82.91** |

## 5 ANALYSIS

### 5.1 ABLATION STUDY

We conduct an ablation study on diverse loss components, as presented in Table 4, utilizing the Tiny-ImageNet and CUB-200 datasets. The initial four rows of the table demonstrate the outcomes obtained by excluding each individual component. The fifth and sixth rows correspond to the cross-entropy and symmetric cross-entropy loss, respectively. Subsequently, the sixth and seventh rows represent the forward and Reverse KL divergence losses. Based on the first and the last row, we can observe that the cross-entropy loss is crucial, and this component's absence results in a failure of network convergence. Removing reverse cross-entropy has the least impact on the performance of the network. However, this results in a non-optimal $Q$-Matrix (Refer to Appendix Figure 9b). We showcase the learned Q matrices for all these discussed scenarios in Appendix H. We can conclude that achieving the network's optimal performance necessitates the inclusion of all loss components.

### 5.2 CLUSTERS VISUALIZATION

We present a visual analysis of clusters formed by 1-hot, Label Smoothing, and Learnable Label Smoothing targets using TSNE (Van der Maaten & Hinton (2008)). Following the experimental setup outlined in (Müller et al. (2019)) for CIFAR-10, we display the penultimate layer features in Figure 5 for all the categories. In the upper row, it is evident that clusters formed by 1-hot targets are dispersed, while those generated by Label Smoothing and Learnable Label Smoothing result in more cohesive and compact clusters. Moving to the second row, we delve into illustrating inter-category relationships by examining distances among cluster centers of the training data. We employed L1-normalized cosine distances, defined as $\frac{cd(i,j)}{\sum_j cd(i,j)}$, where $cd(i, j) = 1 - \frac{c_i \cdot c_j}{||c_i|| \cdot ||c_j||}$, and $c_i$ and $c_j$ represent the cluster centers of categories $i$ and $j$, respectively. Notably, Label Smoothing disrupts the inter-category relationship, rendering all categories equidistant from each other in feature space. In contrast, both 1-hot and Learnable Label Smoothing maintain the inter-category relationship. To further reinforce our findings, we provide more fine-grain visualizations in the Appendix Section E.

To further reinforce our findings, we narrow down the focus to visualize the class-wise distances among select trios from CIFAR-10 and CIFAR-100, mirroring the approach in (Müller et al. (2019)). For these experiments, we concentrated on the *Dog*, *Cat*, and *Truck* classes from CIFAR-10, and the *Beaver*, *Dolphin*, and *Otter* classes from CIFAR-100. The results are showcased in Figure 6, significantly reinforcing our findings. In this figure, we can visualize the distance between the semantically related classes, such as *Cat* and *Dog* in CIFAR-10, or *Beaver* and *Otter* in CIFAR-100 being disrupted by Label Smoothing but remaining intact with Learnable Label Smoothing.

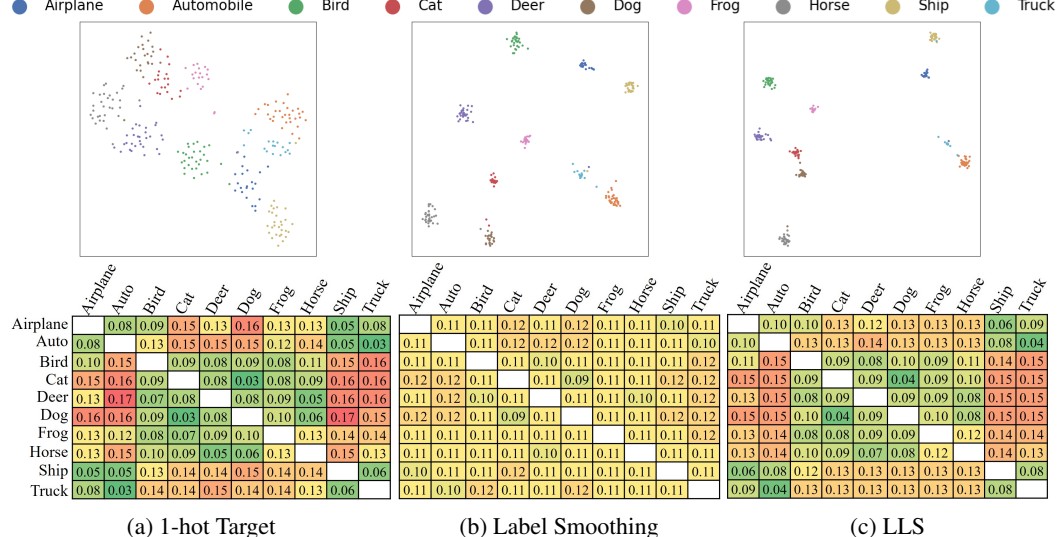

Figure 5: Upper Row: TSNE visualization depicting penultimate features of CIFAR-10. Lower Row: L1 normalized cosine distance among category centers to depict inter-category relationships. In the upper row, it is evident that the category clusters associated with 1-hot targets exhibit dispersion, while those of Label Smoothing and LLS appear more concentrated. In the lower row, it becomes apparent that Label Smoothing disrupts the inter-category relationships, resulting in equal distances between features of all categories. Conversely, 1-hot targets and LLS maintain and have similar inter-category relationships. Our approach provides the advantages of both techniques.

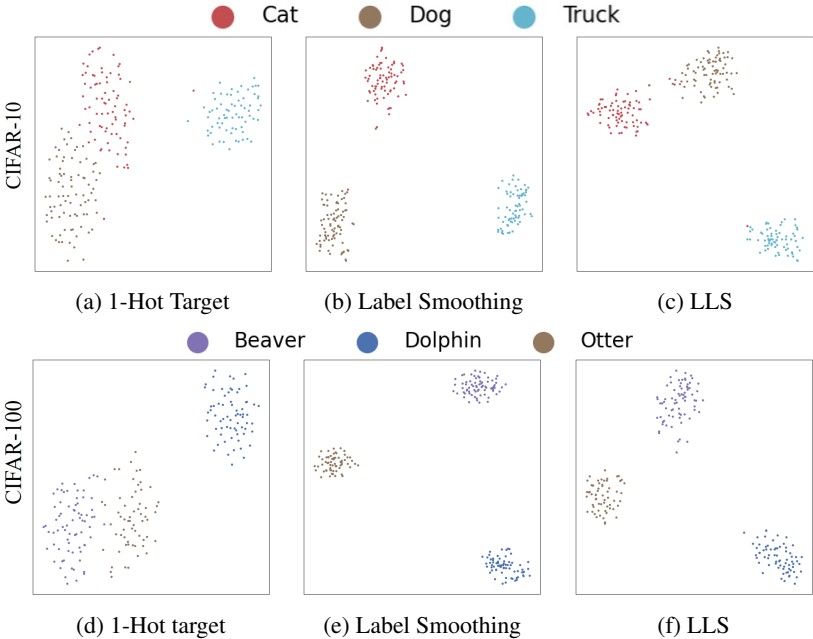

Figure 6: Fine-grain TSNE visualizations illustrating three classes from CIFAR-10 (*Cat*, *Dog*, *Truck*) in the top and CIFAR-100 (*Beaver*, *Dolphin*, *Otter*) in the bottom row. We observe the same behavior as Figure 5. The clusters formed by employing one-hot targets appear scattered whereas label smoothing and LLS result in tightly knit clusters. Furthermore, we can visualize the distance between the semantically related classes, such as *Cat* and *Dog* in CIFAR-10, or *Beaver* and *Otter* in CIFAR-100 being disrupted by Label Smoothing but remaining intact by LLS.

Table 5: Knowledge Distillation experiments. RX: ResNet-X and M2: MobileNetV2. $Y \to Z$ denotes distillation from Y (Teacher) to Z (Student). The rows labeled 1-Hot, LS, and LLS correspond to scenarios where the Teacher network was trained using 1-hot encoding, Label Smoothing, and Learnable Label Smoothing, respectively. For LLS-ST (Learnable Label Smoothing-Substitute Teacher), only the learned $Q$-Matrix from the LLS Teacher network is used for distillation.

| Dataset | CIFAR100 | | | Tiny-ImageNet | | IN100 | CUB200 | | Flowers102 | |
|---|---|---|---|---|---|---|---|---|---|---|
| Teacher | R34 | R34 | R34 | R50 | R101 | R50 | R101 | R101 | R101 | R101 |
| Student | R18 | R34 | R50 | R18 | R18 | R18 | R50 | M2 | R50 | M2 |
| 1-Hot | 78.67 | 79.09 | 80.83 | 63.76 | 63.93 | 83.44 | 81.57 | 78.82 | 92.00 | 91.64 |
| LS | 79.40 | 80.15 | 81.15 | 64.31 | 64.02 | 83.32 | 82.91 | 79.70 | 92.86 | 92.44 |
| LLS | **79.66** | **80.19** | **81.26** | **65.69** | **66.11** | **83.62** | **83.38** | **80.15** | **93.40** | **92.63** |
| LLS-ST | 79.57 | 79.66 | 81.24 | 63.79 | 64.09 | 82.50 | 83.02 | 79.62 | 93.14 | 92.49 |

## 5.3 SUBSTITUTE TEACHER FOR KNOWLEDGE DISTILLATION

Knowledge distillation employs a pre-trained teacher model $f_t$ on dataset to instruct the student model $f_s$. The teacher model generates targets for each sample, which the student model then uses to learn. The training loss for the student network is defined as:

$$\mathcal{L}_{KD} = \beta H(f_s(x), y) + (1 - \beta) H(f_s(x)/T, f_t(x)/T) \tag{6}$$

Here, $H$ represents the cross-entropy loss, $\beta$ is a parameter balancing the use of one-hot labels and teacher targets, and $T$ is the temperature-regulating knowledge transfer from teacher to student. However, the availability of a Teacher network can be constrained by computational or privacy considerations. In such scenarios, the $Q$-matrix of the Teacher network can serve as a substitute Teacher for Knowledge Distillation, denoted as LLS-ST. While a Teacher model furnishes targets on a per-sample basis, LLS-ST exclusively offers category-wise targets only.

Across all datasets, we adopted their original training setup for knowledge distillation but altered the training loss function. Following the recent setup of knowledge distillation experiments, we set $\beta = 0$, implying that student networks are exclusively trained using teacher predictions, and used a temperature of 1 for all experiments. The results are presented in Table 5. The outcomes indicate that networks trained with LLS exhibit superior teaching capabilities during the distillation process. Remarkably, LLS-ST, despite imparting limited knowledge, imparts a performance boost comparable to employing a fully trained Teacher network (refer Table 1 and 2).

## 6 LIMITATIONS

The biggest drawback of our approach is that it requires $K^2$ additional parameters. This becomes a concern when the number of classes grows large, like ImageNet-21k. In such a case, the number of parameters becomes substantially high (441M parameters for ImageNet-21k). To solve this, we propose to merge non-similar categories and keep a fixed number of top similar $N$ during training for each category. The $Q$-Matrix will start with $K \times K$ parameters but will reduce it to $K \times N$ where $N << K$, thereby reducing the number of parameters. We will keep it as part of future exploration work.

## 7 CONCLUSIONS

In our paper, we introduce an innovative label regularization technique named *Learnable Label Smoothing* (LLS). Our approach focuses on empowering networks to learn optimal target labels for regularization. Consequently, our method effectively produces compact feature clusters while preserving the inter-category relationships. Furthermore, the acquired understanding of these inter-category relationships is transferable, aiding in Knowledge Distillation even in scenarios where a Teacher network is unavailable. We believe Learnable Label Smoothing will play a transformative role in knowledge transfer paradigms for neural networks.

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

## A  PYTORCH PSEUDO CODE

```
1  # Define LLS
2  class LLS(nn.Module):
3    def __init__(self, K, alpha=0.1):
4      super().__init__()
5      self.K = K
6      self.alpha = alpha
7      self.qmatrix = nn.Parameter(torch.zeros(K, K), requires_grad=True)
8
9    def forward(self, logits, y):
10     pred = F.softmax(logits, 1)
11
12     y_tgt = (1- α) * F.one_hot(y, num_classes=self.K)
13             + α * F.softmax(self.qmatrix[y], 1)
14
15     forward_kl = KL(y_tgt, pred)
16     backward_kl = KL(pred, y_tgt)
17     loss = (forward_kl + backward_kl)/2
18
19     return loss
20
21  # Define loss function
22  loss_fn = LLS(K, α)
23
24  # Add Q-Matrix parameters to Optimizer
25  params = list(net.parameters()) + list(loss_fn.parameters())
26  optimizer = SGD(params, lr, mom, wd)
```

## B  NECESSITY OF REVERSE KL USING GRADIENT DERIVATION OF LLS

In this section, we present the derivatives of all components comprising our loss function $\mathcal{L}_{lls}$ with respect to the $Q$-Matrix and compare them against the gradient of forward KL for the network. To facilitate the derivations, We employ specific notations: let $q = Q_y = [q_1, q_2, \ldots, q_K]$, where $q_i$ represents the probability of the $i$-th category for the $y$-th row of the $Q$-Matrix. The entries in the $Q$-Matrix are generated from logits. For e.g., $[t_1, t_2, \ldots, t_K]$ are the logits that generate the $y$-th row in $Q$. Here, $q = softmax([t_1, t_2, \ldots, t_K])$, indicating that $q$ is obtained by applying the Softmax activation function to the logits values. Likewise, We use $z = [z_1, z_2, \ldots, z_K]$ to represent the logits from the network $f_\theta$ which are then converted to predicted probabilities $\hat{p}$. Here, $\hat{p} = softmax([z_1, z_2, \ldots, z_K])$.

Firstly, We derive the gradient of the softmax probability $q_i = \frac{e^{t_i}}{\sum_k e^{t_k}}$ with respect to logits $t_j$, as this derivation will be utilized in subsequent derivative calculations,

$$q_i = \frac{e^{t_i}}{\sum_k e^{t_k}}$$
$$\frac{\partial q_i}{\partial t_j} = \frac{e^{t_i} \cdot I\{i = j\}}{\sum_k e^{t_k}} \cdot \frac{\sum_k e^{t_k}}{\sum_k e^{t_k}} - \frac{e^{t_i}}{\sum_k e^{t_k}} \cdot \frac{e^{t_j}}{\sum_k e^{t_k}}$$
$$= q_j I\{i = j\} - q_i q_j$$
$$= q_j (I\{i = j\} - q_i) \tag{7}$$

Similarly, We have the derivative,

$$\frac{\partial \hat{p}_i}{\partial z_j} = \hat{p}_j (I\{i = j\} - \hat{p}_i) \tag{8}$$

We show the derivative of the forward KL loss $\mathcal{L}_{fkl}$ w.r.t. network logits $z_j$:

$$\mathcal{L}_{fkl} = KL(p^{lls}||\hat{p}) = H(p^{lls}, \hat{p}) - H(p^{lls})$$

$$= -\sum_i p_i^{lls} \log \hat{p}_i + \sum_i p_i^{lls} \log p_i^{lls} \tag{9}$$

$$\frac{\partial \mathcal{L}_{fkl}}{\partial z_j} = -\sum_i p_i^{lls} \frac{\partial \log \hat{p}_i}{\partial z_j} + 0$$

$$= -\sum_i \frac{p_i^{lls}}{\hat{p}_i} \frac{\partial \hat{p}_i}{\partial z_j} \quad \text{using Eq.8,}$$

$$= -\sum_i \frac{p_i^{lls}}{\hat{p}_i} \hat{p}_j (I\{i=j\} - \hat{p}_i)$$

$$= -\frac{p_j^{lls}}{\hat{p}_j} \hat{p}_j + \sum_i \frac{p_i^{lls}}{\hat{p}_i} \hat{p}_j \cdot \hat{p}_i$$

$$= -p_j^{lls} + \hat{p}_j \sum_i p_i^{lls}.$$

$$= \hat{p}_j - p_j^{lls} \tag{10}$$

Next, We derive the gradient of forward KL loss $\mathcal{L}_{fkl}$ w.r.t. $Q$-Matrix logits $t_j$:

$$\mathcal{L}_{fkl} = KL(p^{lls}||\hat{p}) = H(p^{lls}, \hat{p}) - H(p^{lls})$$

$$= -\sum_i p_i^{lls} \log \hat{p}_i + \sum_i p_i^{lls} \log p_i^{lls}$$

$$\text{Define} \quad \mathcal{L}_{fce} = -\sum_i p_i^{lls} \log \hat{p}_i \quad \text{and}$$

$$\mathcal{L}_{emt} = \sum_i p_i^{lls} \log p_i^{lls} \tag{11}$$

Next, We will solve derivatives of $\mathcal{L}_{fce}$ and $\mathcal{L}_{emt}$ separately and then combine them later to get the derivative of $\mathcal{L}_{fkl}$.

Derivative of Forward Cross-Entropy loss $\mathcal{L}_{fce}$ w.r.t. $Q$-Matrix logits $t_j$:

$$\mathcal{L}_{fce} = -\sum_i [(1-\alpha)p_i + \alpha q_i] \log \hat{p}_i \tag{12}$$

$$\frac{\partial \mathcal{L}_{fce}}{\partial t_j} = -\sum_i \alpha \frac{\partial q_i}{\partial t_j} \log \hat{p}_i$$

$$= -\alpha \sum_i \frac{\partial q_i}{\partial t_j} \log \hat{p}_i \quad \text{using Eq.7,}$$

$$= -\alpha \sum_i [q_j (I\{i=j\} - q_i)] \cdot \log \hat{p}_i$$

$$= -\alpha q_j \sum_i [(I\{i=j\} - q_i)] \cdot \log \hat{p}_i$$

$$= -\alpha q_j (\log \hat{p}_j - \sum_i q_i \cdot \log \hat{p}_i)$$

$$= \alpha q_j \left( \sum_i q_i \cdot \log \hat{p}_i - \log \hat{p}_j \right) \tag{13}$$

Derivative of Entropy Maximization loss on targets $\mathcal{L}_{emt}$ w.r.t. $Q$-Matrix logits $t_j$:

$$\mathcal{L}_{emt} = \sum_i p_i^{lls} \log p_i^{lls}$$

$$= \sum_i [(1-\alpha)p_i + \alpha q_i] \log[(1-\alpha)p_i + \alpha q_i] \tag{14}$$

$$\frac{\partial \mathcal{L}_{emt}}{\partial t_j} = \sum_i \alpha \frac{\partial q_i}{\partial t_j} \log[(1-\alpha)p_i + \alpha q_i]$$

$$+ \sum_i [(1-\alpha)p_i + \alpha q_i] \cdot \frac{1}{[(1-\alpha)p_i + \alpha q_i]} \cdot \alpha \frac{\partial q_i}{\partial t_j}$$

$$= \sum_i \alpha \frac{\partial q_i}{\partial t_j} \log[(1-\alpha)p_i + \alpha q_i] + \cdot \alpha \sum_i \frac{\partial q_i}{\partial t_j}$$

$$= \alpha \sum_i \frac{\partial q_i}{\partial t_j} \{1 + \log[(1-\alpha)p_i + \alpha q_i]\}$$

$$= \alpha \sum_i (1 + \log p_i^{lls}) \frac{\partial q_i}{\partial t_j} \quad \text{using Eq.7,}$$

$$= \alpha \sum_i (1 + \log p_i^{lls}) \cdot q_j (I\{i=j\} - q_i)$$

$$= \alpha q_j \sum_i (1 + \log p_i^{lls})(I\{i=j\} - q_i)$$

$$= \alpha q_j \left[ (1 + \log p_j^{lls}) - \sum_i q_i (1 + \log p_i^{lls}) \right]$$

$$= \alpha q_j \left[ (1 + \log p_j^{lls}) - \sum_i q_i - \sum_i q_i \log p_i^{lls} \right]$$

$$= \alpha q_j \left[ 1 + \log p_j^{lls} - 1 - \sum_i q_i \log p_i^{lls} \right]$$

$$= \alpha q_j \left[ \log p_j^{lls} - \sum_i q_i \log p_i^{lls} \right]$$

$$= \alpha q_j \left( \log p_j^{lls} - \sum_i q_i \log p_i^{lls} \right) \tag{15}$$

The Final derivative of Forward KL $\mathcal{L}_{fkl}$ w.r.t. $t_j$ can be obtained as:

$$\frac{\partial \mathcal{L}_{fkl}}{\partial t_j} = \frac{\partial \mathcal{L}_{fce}}{\partial t_j} + \frac{\partial \mathcal{L}_{emt}}{\partial t_j} \quad \text{using Eq.13, \& Eq.15,}$$

$$= \alpha q_j \left( \sum_i q_i \cdot \log \hat{p}_i - \log \hat{p}_j \right)$$

$$+ \alpha q_j \left( \log p_j^{lls} - \sum_i q_i \log p_i^{lls} \right)$$

$$= \alpha q_j \left( \sum_i q_i \cdot \log \frac{\hat{p}_i}{p_i^{lls}} - \log \frac{\hat{p}_j}{p_j^{lls}} \right) \tag{16}$$

It can be observed that there is a notable disparity in the gradient of forward KL with respect to $t_j$ as seen in Eq. 16, which is consistently one to two orders of magnitude smaller compared to its counterpart concerning $z_j$ in Eq. 10. This discrepancy arises due to the logarithmic scaling effect on the gradients, resulting in a reduction in magnitude. To enhance the flow of gradients into the $Q$-Matrix without resorting to increasing the learning rate, we incorporate reverse KL $\mathcal{L}_{rkl}$ in the training loss function. Next, we show the gradient of the reverse KL $\mathcal{L}_{rkl}$ w.r.t. logits $t_j$ of $Q$-Matrix to understand its impact.

The gradient of reverse KL $\mathcal{L}_{rkl}$ w.r.t. logits $t_j$ of $Q$-Matrix can be derived as:

$$
\begin{aligned}
\mathcal{L}_{rkl} &= KL(\hat{p}||p^{lls}) = H(\hat{p}, p^{lls}) - H(\hat{p}) \\
&= -\sum_i \hat{p}_i \log p_i^{lls} + \sum_i \hat{p}_i \log \hat{p}_i \\
&= -\sum_i \hat{p}_i \log[(1-\alpha)p_i + \alpha q_i] + \sum_i \hat{p}_i \log \hat{p}_i \quad (17)
\end{aligned}
$$

$$
\begin{aligned}
\frac{\partial \mathcal{L}_{rkl}}{\partial t_j} &= -\sum_i \frac{\hat{p}_i}{(1-\alpha)p_i + \alpha q_i} \cdot \alpha \frac{\partial q_i}{\partial t_j} + 0 \\
&= -\alpha \sum_i \frac{\hat{p}_i}{(1-\alpha)p_i + \alpha q_i} \cdot \frac{\partial q_i}{\partial t_j} \quad \text{using Eq.7,} \\
&= -\alpha \sum_i \frac{\hat{p}_i}{(1-\alpha)p_i + \alpha q_i} \cdot q_j(I\{i=j\} - q_i) \\
&= -\alpha q_j \sum_i \frac{\hat{p}_i}{(1-\alpha)p_i + \alpha q_i} \cdot (I\{i=j\} - q_i) \\
&= -\alpha q_j \left[ \frac{\hat{p}_j}{(1-\alpha)p_j + \alpha q_j} - \sum_i \frac{\hat{p}_i \cdot q_i}{(1-\alpha)p_i + \alpha q_i} \right] \\
&= -\alpha q_j \left[ \frac{\hat{p}_j}{p_j^{lls}} - \sum_i \frac{\hat{p}_i \cdot q_i}{p_i^{lls}} \right] \\
&= \alpha q_j \left( \sum_i q_i \cdot \frac{\hat{p}_i}{p_i^{lls}} - \frac{\hat{p}_j}{p_j^{lls}} \right) \quad (18)
\end{aligned}
$$

By examining Eq. 16 and 18, it becomes apparent that the gradients of forward and reverse KL exhibit strong similarities, differing primarily due to the presence of the $\log$ function in the $\frac{\hat{p}_i}{p_i^{lls}}$ terms. $\log$ function diminished the gradient values in the forward KL scenario, whereas, in reverse KL, the unscaled values are employed. Interestingly, the gradients derived from reverse KL align in magnitude order with those of forward KL concerning $z$ in Eq 10. This leads to a better convergence of the $Q$-Matrix.

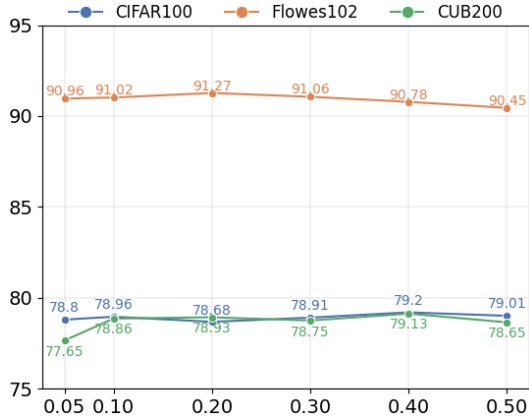

Figure 7: Results on varying $\alpha$ on CIFAR100, Flowers-102, and, CUB200 dataset with ResNet-18. We can observe that $\alpha \in (0.1, 0.4)$ provides the best overall performance.

Table 6: The comparison between applying the learned $Q$-Matrix from the full data vs. employing 1-hot encoding, Label Smoothing, and Learnable Label Smoothing on sample-wise subsets. The results demonstrate a significant boost in generalization when the learned $Q$-Matrix is applied to the sample-wise subsets.

|  | ImageNet100 | TinyImageNet | FMNIST | CIFAR100 |
|---|---|---|---|---|
| 1-Hot | 77.22 | 54.26 | 86.49 | 73.70 |
| LS | 78.62 | 54.70 | 87.01 | 74.56 |
| LLS | 78.66 | 54.85 | 87.13 | 74.75 |
| LLS-ST | **79.0**2 | **55.21** | **87.56** | **74.92** |

## C  VARYING HYPERPARAMETER $\alpha$

We show outcomes obtained by varying the values of hyperparameter $\alpha$ (0.05, 0.1, 0.2, 0.3, 0.4, and 0.5) using a ResNet18 on CIFAR-100, Flowers-102, and CUB-200 datasets in Figure 7. Our results indicate that the range $\alpha \in (0.1, 0.4)$ consistently delivers the optimal performance across these datasets.

## D  EFFECTIVENESS ON SUBSETS OF DATA

It's expected that the $Q$-Matrix is predominantly shaped by the characteristics of the training data, and any alterations to the training dataset consequently influence the learned $Q$-Matrix. However, once the $Q$-Matrix has been acquired, it remains applicable to both its category-wise and sample-wise subsets.

In this experiment, we meticulously examine these two types of subsets: (1) selecting the first 50% of categories and (2) randomly choosing 50% of samples from ImageNet-100, TinyImageNet, FashionMNIST, and CIFAR-100 datasets. Employing these data subsets, we train a ResNet-18 with 1-hot targets, Label Smoothing, and Learnable Label Smoothing as baselines. Subsequently, we delve into the impact of applying the learned $Q$-Matrix from the entire dataset, similar to the substitute Teacher experiments (LLS-ST). For category subsets, we extract the logits corresponding to the selected categories from the $Q$-Matrix and exclusively apply Softmax to these chosen values.

The results of these experiments are detailed in Table 6 and 7. Notably, the learned $Q$-Matrix exhibits superior performance when applied to a subset of samples. When dealing with a subset of categories, learning a new $Q$-Matrix enhances generalization, with the learned matrix closely approaching the performance of a newly trained matrix, outperforming 1-hot targets and Label Smoothing.

Table 7: The comparison between applying the learned $Q$-Matrix from the full data vs. employing 1-hot encoding, Label Smoothing, and Learnable Label Smoothing on class-wise subsets. When working with a subset of categories, acquiring a new $Q$-Matrix results in superior performance. However, the learned $Q$-matrix demonstrates a close alignment with the performance of a freshly trained $Q$-matrix.

|        | ImageNet100 | TinyImageNet | FMNIST | CIFAR100 |
|--------|-------------|--------------|--------|----------|
| 1-Hot  | 76.68       | 66.90        | 89.86  | 83.24    |
| LS     | 77.88       | 66.98        | 90.56  | 83.56    |
| LLS    | **78.56**   | **67.48**    | **91.12** | **83.66** |
| LLS-ST | 78.20       | 67.08        | 90.96  | 83.64    |

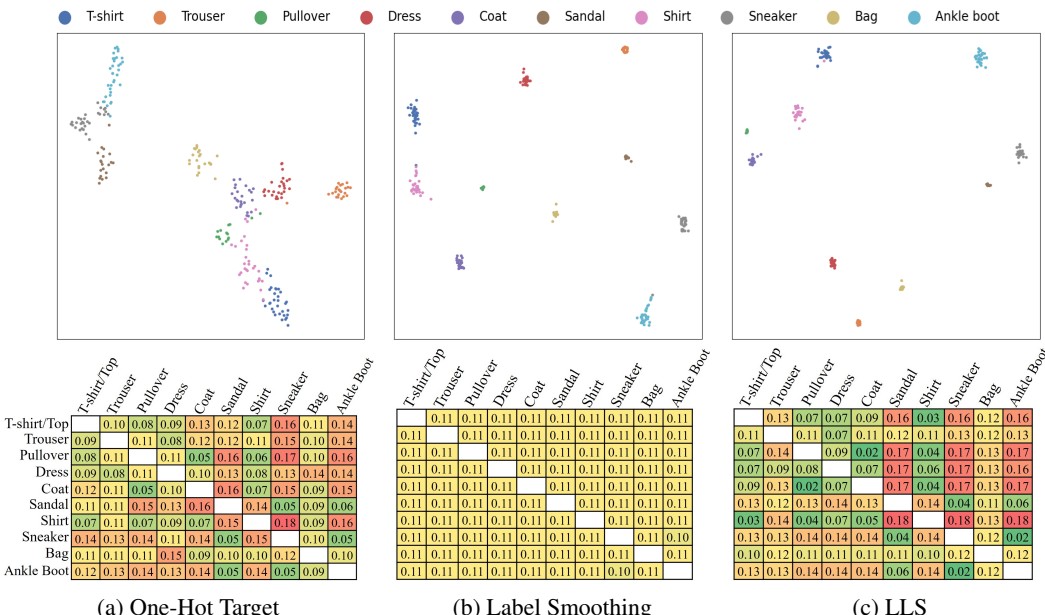

(a) One-Hot Target     (b) Label Smoothing     (c) LLS

Figure 8: Upper Row: TSNE visualization depicting penultimate features of FashionMNIST. Lower Row: L1 normalized cosine distance between class cluster centers. In the upper row, it is evident that the class clusters associated with one-hot targets exhibit dispersion, while those of Label Smoothing and Learnable Label Smoothing appear more concentrated. Moving to the lower row, it becomes apparent that label smoothing disrupts the inter-class relationships, resulting in equal distances between all classes. Conversely, one-hot targets and Learnable Label Smoothing maintain and preserve these inter-class relationships. Notably, Learnable Label Smoothing combines the advantages of both techniques.

# E  MORE CLUSTER VISUALIZATIONS

We replicated the experiment outlined in the cluster visualization section of the main text using the FashionMNIST dataset with a LeNet architecture. The outcomes of these experiments are presented in Figure 8. In the top row of visualizations, we visualize the features using TSNE. In the bottom row, we calculated class cluster centers using the training data and presented L1-normalized cosine distances among classes as $\frac{cd(i,j)}{\sum_j cd(i,j)}$, where $cd = 1 - \frac{c_i \cdot c_j}{||c_i|| \cdot ||c_j||}$, and $c_i$ and $c_j$ denote the cluster centers of class $i$ and $j$, respectively.

These comparisons underline that clusters formed by 1-Hot targets demonstrate dispersion, while those formed by label smoothing and learnable label smoothing exhibit a more cohesive and compact nature. Notably, label smoothing consistently disrupts inter-class relationships, equidistantly positioning classes within the feature space—a salient observation underscored in our findings. In

Table 8: Application of Label Smoothing++ with Input Augmentations techniques - Co: Cutout, Mx: Mixup, Cx: CutMix, RA: RandAugment.

| Dataset | Animals-10N | ImageNet-100 |
|---|---|---|
| 1-hot | 85.00 | 81.72 |
| LLS | 86.69$_\uparrow$ | 82.72$_\uparrow$ |
| Co (Devries & Taylor (2017)) | 86.80 | 82.86 |
| Co + LLS | 88.04$_\uparrow$ | 82.96$_\uparrow$ |
| Mx (Zhang et al. (2018)) | 87.37 | 81.88 |
| Mx + LLS | 87.50$_\uparrow$ | 82.48$_\uparrow$ |
| Cx (Yun et al. (2019)) | 88.00 | 83.50 |
| Cx + LLS | 88.58$_\uparrow$ | 83.68$_\uparrow$ |
| RA (Cubuk et al. (2020)) | 86.62 | 82.88 |
| RA + LLS | 87.24$_\uparrow$ | 83.50$_\uparrow$ |

Table 9: Comparison of the number of training parameters and training time on Tiny-ImageNet with ResNet-18 and ResNet-101.

| | ResNet-18 | | ResNet-101 | |
|---|---|---|---|---|
| | Parameters | Time (mins) | Parameters | Time (mins) |
| 1-hot | 11,578,632 | 142 | 44,131,080 | 674 |
| LS | 11,578,632 | 142 | 44,131,080 | 674 |
| LLS | 11,618,632 (0.3%$\uparrow$) | 146 (1.4%$\uparrow$) | 44,171,080 (0.1%$\uparrow$) | 680 (0.89%$\uparrow$) |

contrast, both One-hot encoding and learnable label smoothing methods consistently uphold and sustain inter-class relationships effectively. Significantly, learnable label smoothing emerges clearly superior by showcasing the strengths of both methods.

## F  COMPATIBILITY WITH INPUT AUGMENTATIONS

In this section, we assess the compatibility of our approach with input augmentation techniques such as Cutout, Mixup, Cutmix, and Randaugment. The results of this experiment are presented in Table 8 using CIFAR100, FashionMNIST, Tiny-ImageNet, and ImageNet-100 datasets with ResNet34, ResNet18, ResNet18, and ResNet18, respectively. Our findings indicate that label regularization seamlessly integrates with input regularization techniques. Employing input and label regularization together yields optimal performance, as evidenced by the results in the table.

## G  COMPUTATION OVERHEAD OF LLS

We examine the computation overhead of learnable label smoothing. LLS adds $K^2$ extra parameters which scales quadrically with the number of classes. Hence, we use the Tiny-ImageNet dataset as it has the highest number of classes (200) in our experiments. We show the total number of trainable parameters and training time with ResNet-18 and ResNet-101 in table 9. As per the results, Learnable Label Smoothing adds less than 0.3% parameters in both cases and increases training time by about 1%.

## H  Q-MATRICES FROM ABLATION STUDY

In the main text, we conducted an ablation study to assess and compare their outcomes in terms of performance. In this section, we present the learned Q-Matrices on CIFAR-10 for the same experiments. The outcomes are illustrated in Figure 9. Notably, it's apparent that the $Q$-Matrix achieves its

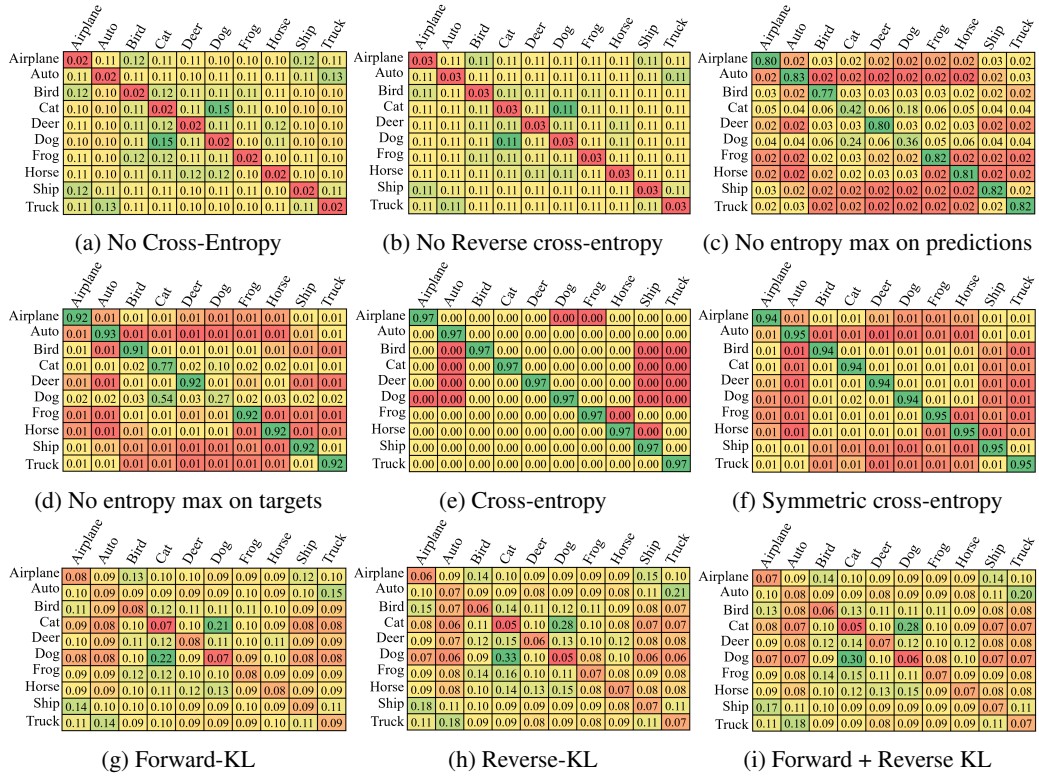

Figure 9: Learned $Q$-Matrix on CIFAR-10 as per the ablation study experiments.

optimal state exclusively with backward KL and forward-backward loss functions. Conversely, employing the Forward KL approach results in slower convergence (as shown in Section B), ultimately leading to suboptimal values.

## I TRAINING SETUP

### I.1 DATASET DETAILS

**CIFAR10 and CIFAR100**

- Augmentations: Utilized padding of size 4, Random Crops, and random horizontal flips during training.
- Optimizer: Employed SGD optimizer with 0.9 momentum and weight decay of 5e-4.
- Training specifics: Networks were trained with a batch size of 128 for 300 epochs. The learning rate initiated at 0.1 and warmed up linearly for the first 10 epochs. Then, it decayed by a factor of 0.1 at the 150th and 225th epochs.

**FashionMNIST**

- Augmentation: Applied padding of size 2 with random crops as the sole augmentation.
- Network Configuration: Set the input channels to 1 for grayscale images.
- Optimizer: Employed SGD optimizer with 0.9 momentum and weight decay of 1e-4.
- Training specifics: Networks were trained with a batch size of 128 for 200 epochs. The learning rate began at 0.1, underwent a linear warm-up for the initial 5 epochs, and decayed by a factor of 0.1 at the 100th and 150th epochs.

**SVHN**

- No augmentation was used for this dataset.

- Optimizer: Used SGD optimizer with 0.9 momentum and weight decay of 1e-4.

- Training specifics: Networks were trained with a batch size of 128 for 200 epochs. The learning rate started at 0.1, had a linear warm-up for the first 5 epochs, and decayed by a factor of 0.1 at the 100th and 150th epochs.

**TinyImageNet**

- Image size: Images in TinyImageNet data were of size $64 \times 64$.

- Augmentations: Implemented padding of size 4, Random Crops, and random Horizontal flips.

- Optimizer: Used SGD optimizer with 0.9 momentum and weight decay of 1e-4.

- Training specifics: Networks were trained with a batch size of 64 for 100 epochs. The learning rate began at 0.1, underwent a linear warm-up for the first 5 epochs, and decayed by a factor of 0.1 at the 40th and 60th epochs.

**Animals10N**

- Image size: Images in Animals10N data were of size $64 \times 64$.

- Augmentations: Implemented padding of size 4, Random Crops, and random Horizontal flips.

- Optimizer: Used SGD optimizer with 0.9 momentum and weight decay of 1e-4.

- Training specifics: Networks were trained with a batch size of 64 for 100 epochs. The learning rate began at 0.1, underwent a linear warm-up for the first 5 epochs, and decayed by a factor of 0.1 at the 40th and 60th epochs.

**ImageNet-100**

- Image size: Training images were of the original ImageNet dataset size $224 \times 224$.

- Augmentations: Employed (1) Standard augmentation of random resized crops of 224 along with random Horizontal flips. (2) Standard augmentation with RandAugmentation.

- Optimizer: Utilized SGD optimizer with 0.9 momentum and weight decay of 1e-4.

- Training specifics: Networks were trained with a batch size of 64 for 90 epochs. The learning rate began at 0.1, underwent a linear warm-up for the first 5 epochs, and decayed by a factor of 0.1 at the 30th, 60th, and 80th epochs.

**CUB200 and Flowers102**

- Approach: Utilized pretrained networks for these datasets, adapting the last classification layer based on the dataset's class count.

- Augmentation: Images were scaled to 256 and then randomly cropped to 224 for augmentation, along with random horizontal flips.

- Optimizer: Employed SGD optimizer with 0.9 momentum and weight decay of 1e-4.

- Training specifics: Networks were trained with a batch size of 64 for 100 epochs. The learning rate initiated at 0.01, underwent a linear warm-up for the first 5 epochs, and decayed by a factor of 0.1 at the 45th and 80th epochs.

