# OpenReview forum: "LLS: Regulating Neural Network Training via Learnable Label Smoothing"
_ICLR.cc/2025/Conference — ICLR 2025 Conference Withdrawn Submission_

### Official Review · Reviewer_y2XN · 2024-10-27

**Soundness:** 3
**Presentation:** 3
**Contribution:** 3
**Rating:** 5
**Confidence:** 4

**Summary:**

This paper proposes a learnable label smoothing method that utilizes the relationships among categories rather than the uniform distribution in the well-known label smoothing. The motivation is clear, and the method is simple. The paper shows the learned labels and demonstrates that the label can learn similarities between categories, which verifies the motivation.

**Strengths:**

This paper proposes a learnable label smoothing method that utilizes the relationships among categories rather than the uniform distribution in the well-known label smoothing. The motivation is clear, and the method is simple. The paper shows the learned labels and demonstrates that the label can learn similarities between categories, which verifies the motivation.

**Weaknesses:**

- The proposed LLS brings marginal gain over other variants, like well-known LS, TFKD, and OLS. For example, In Table 3, on the ImageNet-100 dataset, the proposed LLS (84.90) shows similar performance as TFKD (84.72) and OLS (84.71).
- The experiments in this paper are insufficient. As a general regularization method for classification, this paper is expected to provide experimental results on the ImageNet-1K with ResNet-50 or a transformer-based backbone at least.

**Questions:**

- The first loss term in the method is the KL divergence between the learned label and the prediction distribution. In the early training epochs, the prediction tends to be misclassified. So I wonder if the noisy prediction logits harm the learned labels, and how to deal with this case.
- It is suggested to report the performance of LLS on the ImageNet-1K dataset or some downstream tasks, such as semantic segmentation and object detection.
- The learned labels are required to learn the prediction logits and the labels are shared by the whole dataset. What is the superiority of the proposed learned labels compared to the OLS with statistical labels? Is there a theoretical or intuitive explanation?
- How to deal with cases that use binary cross-entropy classification loss?

---

### Official Review · Reviewer_Y23e · 2024-10-28

**Soundness:** 3
**Presentation:** 4
**Contribution:** 3
**Rating:** 6
**Confidence:** 5

**Summary:**

This paper proposes a novel Learnable Label Smoothing (LLS) to regularize the supervised learning. Specifically, a learnable Q-Matrix is utilized to optimize label encoding for each category, thereby effectively modeling the relationships across distinct classes compared to vanilla Label Smoothing (LS). Experimental results verify LLS outperforms LS and its variants. Also, the learned Q-matrix can serve as a Teacher model for knowledge distillation.

**Strengths:**

1.	The paper is well-written, and LLS is simple yet effective.

2.	Extensive experimental results demonstrate the effectiveness of LLS.

3.	The learned Q-matrix can serve as a substitute Teacher for Knowledge distillation, which is particularly useful in privacy-protection scenarios.

**Weaknesses:**

1.	Could the optimal Q-matrix be theoretically symmetric? Would it be feasible to introduce symmetry constraints during the learning process, or alternatively, to learn only half of the matrix (e.g., the upper triangular portion)?


2.	Is there a theoretical relationship between $\mathcal{L}_{lls}$ and the vanilla cross-entropy loss?

If the above issues are adequately addressed, I will consider increasing my score further.

Small issues:

1.	The citation format used in this paper appears incorrect, please refine it carefully.

2.	Line 214-215: $-H(p_{lls}, \hat{p})$ - > $H(p_{lls}, \hat{p})$,  $-H(\hat{p}, p_{lls})$ - > $H(\hat{p}, p_{lls})$

**Questions:**

See weakness for details.

---

### Official Review · Reviewer_Biqo · 2024-11-05

**Soundness:** 3
**Presentation:** 2
**Contribution:** 3
**Rating:** 6
**Confidence:** 4

**Summary:**

This paper proposes a novel Learnable Label Smoothing framework, which could learn extra information in order to preserve inter-category relationships through the learning procedure.  Extensive experiments on fine-grained classification and knowledge distillation experiments demonstrate the effectiveness of the proposed framework.

**Strengths:**

+ The proposed learnable label smoothing framework is well-designed and could be utilized as a general approach to preserve inter-category relationships.
+ Experiments on two different domains (i.e., fine-grained classification and knowledge distillation experiments) demonstrate the generality and effectiveness of the proposed method.

**Weaknesses:**

- Though the experiments on knowledge distillation show the effectiveness of the proposed method, the reason why it could work well is not clearly presented and discussed in the paper. As a comparison, for the fine-grained classification task, the paper demonstrates a toy diagram (Figure 1) and discusses the learnable label smoothing (LLS) can help preserve the inter-category relationship in detail. It is very helpful to understand why LLS works for this task. For the knowledge distillation task, could you please also more illustrations or discussions about how and why LLS can work on the knowledge distillation task?

**Questions:**

- In the “Limitation” section, it mentions the number of additional parameters becomes substantially high when the number of classes grows large. It may cause unfair comparisons with the same backbone approach because the proposed method could have a large number of additional parameters to learn more information. Could the author provide more information to compare the proposed method to the baseline methods with the same amount of parameters?

---

### Official Review · Reviewer_aqmP · 2024-11-06

**Soundness:** 2
**Presentation:** 3
**Contribution:** 2
**Rating:** 3
**Confidence:** 4

**Summary:**

The authors propose to replace the uniform probability vector of Label Smoothing with a learnable Q-matrix, which leads to unique targets for each category. The proposed method is evaluated on several image classification datasets and shows improved performance against Label Smoothing and its variants.

**Strengths:**

- The idea seems to be reasonable and the method is clearly described.
- The paper is well written and easy to follow.
- Comprehensive experiments are conducted across datasets and network architectures.

**Weaknesses:**

- I have a great concern on the reported experiment results of other methods, which are indeed not directly comparable due to their different baseline accuracies. For example, I could confirm that the reported results of 1-Hot, LS and OLS on CIFAR 100 in Table 2 are directly cited from OLS paper, but these are the average results of three runs. How could LLS be directly compared to them without reporting its own baseline accuracy, since there are several popular resnet repositories with different training setups? Similarly, Zipf-LS reported an average accuracy of 77.38 with ResNet-18, but their reported average baseline accuracy is 75.51, which is much lower than the listed baseline accuracy of 75.87 in Table 2. These problems indicate that the results in Table 1 and Table 2 are indeed not directly comparable. Assuming the reported results for LLS is the average accuracy, then the standard deviation is also missing.

- Despite that the proposed method is evaluated on many small datasets,  a larger dataset, such as the most representative ImageNet-1K, is not included.

- Theoretically speaking, It is unclear why directly optimising the parameters of the network and the parameters of the Q-matrix simultaneously would bring such a benefit to the classification. Although it might converge to a minimum loss with respect to both parameters, it doesn't necessarily imply that the inter-category relationship is encoded in the Q-matrix, because there's a high chance of learning some collapsed representation irrelevant to classification. For similar cases in prior works [1] [2], stop gradient operation is often adopted to force updating only the parameters of one side for different loss terms, instead of optimising both sides jointly. However, the lack of theoretical support is not a major weak point and my decision is mainly based on whether the experiment results are convincing or not.

[1] Van Den Oord, Aaron, and Oriol Vinyals. "Neural discrete representation learning." Advances in neural information processing systems 30 (2017).

[2] Grill, Jean-Bastien, et al. "Bootstrap your own latent-a new approach to self-supervised learning." Advances in neural information processing systems 33 (2020): 21271-21284.

**Questions:**

- the authors are suggested to report their own reproduced results for other methods in Table 1 and Table 2 to guarantee a valid comparison, please also include the standard deviation if the reported results of LLS is the average of multiple runs.
-  It is important to see whether the proposed method scales up on a larger dataset such as ImageNet-1K.

I would like to raise my score if the results are convincing, regardless of point 3 in weakness.

---

### Note · Authors · 2025-01-16

I have read and agree with the venue's withdrawal policy on behalf of myself and my co-authors.